# Delayed presentation of breast cancer patients and contributing factors in East Africa: Systematic review and meta-analysis

**Chalie Mulugeta**[1]*, **Tadele Emagneneh**[1], **Getinet Kumie**[2], **Betelhem Ejigu**[1], **Abebaw Alamrew**[1]

**1** Department of Midwifery, College of Health Science, Woldia University, Woldia City, Ethiopia,
**2** Department of Medical Laboratory Science, College of Health science, Woldia University, Woldia City, Ethiopia

\* chaliemulu19@gmail.com

**Data Availability Statement:** All relevant data are within the manuscript and its Supporting Information files.

## Abstract

### Introduction

Breast cancer remains a significant public health issue, with delayed medical attention often leading to advanced stages and poorer survival rates. In East Africa, evidence on the prevalence and factors contributing to the delayed presentation of breast cancer is limited. As a result, this study aims to assess the pooled prevalence of delayed breast cancer presentation and identify contributing factors in East Africa.

### Methods

We conducted a systematic review of observational studies from East Africa using PubMed, Google Scholar, Embase, Cochrane Library, Hinari, and Mednar databases. The Newcastle Ottawa 2016 Critical Appraisal Checklist assessed methodological quality. Publication bias was evaluated using a funnel plot and Egger's test, and heterogeneity was examined with the I-squared test. Data were extracted with Microsoft Excel and analyzed using Stata 11.

### Results

The pooled prevalence of delayed presentation among breast cancer patients in East Africa was 61.85% (95% Confidence Interval: 48.83%–74.88%). Significant factors contributing to delayed presentation included visiting traditional healers (Adjusted Odds Ratio: 3.52; 95% CI: 1.43–5.59), low educational levels (Adjusted Odds Ratio: 3.61; 95% CI: 2.39–4.82), age>40 years (Adjusted Odds Ratio 1.87; 1.03, 2.71), absence of breast pain (Adjusted Odds Ratio 2.42; 1.09, 3.74), distance >5km away from home to health institution (Adjusted Odds Ratio 2.89; 1.54, 4.24), and rural residence (Adjusted Odds Ratio: 3.33; 95% CI: 2.16–4.49).

### Conclusion

This meta-analysis's findings demonstrated that over half of breast cancer patients in East Africa delayed detection. Significant factors associated with delayed presentation include

**Funding:** The author(s) received no specific funding for this work.

**Competing interests:** The authors have declared that no competing interests exist.

age over 40 years, illiteracy, rural residence, use of traditional healers, distance greater than 5 km from a health facility, and absence of breast pain. Healthcare stakeholders and policy-makers must be focused on raising awareness and educating people to encourage early detection and prompt therapy.

## Introduction

Globally, 2.3 million women received a breast cancer diagnosis in 2022, and 670,000 people died from the disease [1]. While breast cancer mortality is highest in less developed nations, the disease's incidence is higher in more developed nation [2]. Sub-Saharan Africa as a whole is facing a growing cancer-related public health burden. Currently, 4% of Ethiopian mortalities are related to cancer [3]. In Africa, women die from breast cancer at a rate of 20% and account for 28% of all cancer cases. Incidence rates are still generally varied in Africa, estimated below 35 per 100,000 women in most countries [4], Kenya 52 [5], Zimbabwe 33 [6] and Uganda 34 [7] per 100, 000 women were breast cancer incidence.

Today, the World Health Organization announced the Global Breast Cancer Initiative, a significant new cooperative effort aimed at preventing an estimated 2.5 million deaths world-wide from breast cancer by 2.5% year until 2040 [8]. To achieve these goals, WHO launched early detection (60 percent of cases detected in the early stages), prompt diagnosis (60 days), and thorough care (80% of cases completed with treatment) [9].A systematic review and meta-analysis revealed that the average duration between recognizing symptoms and presenting them to a medical professional was less than 4 months in North Africa and between 3 and 6 months in sub-Saharan Africa [10]. Similar research conducted in Africa showed that the eastern and central areas had the worst rates of late presentation (>90 days), with an overall esti-mate of 54% [11]. A systematic review and meta-analysis done on breast cancer patients report that delays of three to six months are linked to a decreased chance of survival [12].

Evidence showed that the effects of delay on prognosis have generally demonstrated that longer delays are linked to malignancies that are diagnosed at an advanced stage, which lowers the likelihood of survival [12–14]. Longer patient delays were linked to bigger tumor sizes, pos-itive nodes, and a 24% death rate compared to shorter patient delays [15]. Longer delays were associated with lower survival rates for women, both from the date of diagnosis and from the beginning of symptoms [16]. Research done on the types and timing of symptoms experienced by breast cancer patients, the disease does not present with a lump at first present with symp-toms later, and are more likely to have their doctor delayed sending them for a second opinion [17]. Previous Systematic Review and meta-analysis revealed that low education level [10, 18–21], low-income status [10, 18, 21], Symptom misinterpretation [11, 22], preference for alter-native care [11, 21–23], older age [19, 21], no family history of breast cancer [19], not perform-ing breast self -examination [20, 23], not married [21], poor knowledge about cancer [21, 23], socio-cultural factors such as belief [22, 23] and lack of trust in access health care [22] were the contributing factors of late patient presentation of breast cancer.

According to a comprehensive analysis conducted on Asians, breast exams and symptoms have consistently demonstrated a major impact on minimizing the amount of time that a diag-nosis is delayed [24].A systematic review done on barriers of late presentation and late-stage diagnosis of breast cancer revealed that poor awareness of symptoms and risk factors, anxiety about finding breast abnormalities, fear of cancer treatments, worry of partner desertion, shame about revealing symptoms to medical experts, taboo and stigmatism were some of the

variables that contributed to patient delays [25].Previous primary studies were done in the world the coverage of late patient presentation of breast cancer ranges from25% [26]- 89% [27] and positively associated with by range of factors such as individual level (socio-demographic [17, 28, 29], cultural belief [30–32], husband's attitude and support [27, 32], family income [33]), health service level (distance, accessibility, and availability) [29, 30, 34] and knowledge level [27, 31, 34], absence of pain in the breast [35], no family history of breast CA [29] and not practice self-breast examination [29].

There was previous primary research conducted in East Africa to determine the prevalence of delayed presentation of breast cancer patients and associated factors; however, findings from those studies varied across countries. To the best of our knowledge, this topic has not yet been investigated by systematic review and meta-analysis at the regional level. In particular, this study covered a wider geographical area and provided pooled results. This information is necessary for policy planners and program managers to identify gaps in the delayed presentation of breast cancer patients and to plan strategies to reduce the delay of breast cancer patients. The development of successful programs that increase medical seeking consultation enhances survival rates and decreases mortality and morbidity of breast cancer patients in the East Africa Region depends on the identification of a single number of common factors. Early identification and prompt treatment of breast cancer are crucial for improving maternal health. Systematic reviews and meta-analyses are necessary to address this issue. Thus, the goal of this study was to assess the pooled prevalence of delayed presentation of breast cancer patients and contributing factors in East Africa.

## Methods and materials

### Study protocol and reporting

This systematic review and meta-analysis was carried out per the Preferred Reporting Items for Systematic Reviews and Meta-Analyses (PRISMA) criteria [36] (S1 File). The eligibility criteria were adapted from the Newcastle Ottawa 2016 review guidelines [37].We used Endnote (version X7) reference management software to download, organize, and review and Zotero to cite related articles.

### Inclusion criteria

**Study area**: Only research carried out in East Africa.

**Participants in the study**: All quantitative studies with indicators or variables indicating late patient presentation of breast cancer.

**Study types**: Observational cross-sectional studies.

**Results of interest**: The main investigations revealed the frequency of delayed patient presentation and contributing factors.

**Publication condition**: We included published articles written in English. There are no restrictions on race and publication date.

**Language**: Correspondingly, all primary studies published in the English language and reported the prevalence and/or associated factors on Delayed presentation of breast cancer patients in East Africa and fulfill the following criteria were included in this review (Table 1).

**Exclusion criteria.** Excluded from the study were anonymous reports, duplicate research, articles lacking an abstract or full text, and qualitative investigations. We excluded systematic reviews, case reports, and retrospective reviews. We also excluded studies focusing on specific factors and frequency with descriptive studies. Since there was no concrete data to take from this research, they were eliminated. To increase the similarity of the studies included in the

**Table 1. Show the inclusion and exclusion criteria.**

| Study characteristics | Inclusion criteria | Exclusion criteria |
|---|---|---|
| Design | observational studies Cross-sectional studies | Clinical trials, qualitative studies, editorial letters, case reports/series |
| Population | Breast cancer patients | Patients with malignancies of other body parts |
| Condition | Delayed presentation of breast cancer patients | Unclear to determine the time of presentation of breast cancer patient, articles only reviews and descriptive static's |
| Context | Studies conducted in East Africa | Studies not from East Africa |
| Language | English | English |

meta-analysis with regard to all significant factors, research carried out in particular populations was also eliminated.

## Variable measure

Patient delay was defined as time intervals of more than 12 weeks from the first symptom recognition to the first medical consultation [26, 28, 31, 34].

The place of residence was classified as rural or urban and educational status was classified as secondary or above and below secondary, traditional healer was categorized as visit or not visit traditional healer, having no family history of breast cancer was categorized as having a history or not, lump under armpit grouped in to yes or no marital status grouped in to married or not married. Age is grouped into two categories: $< 40$ or $= >40$ years. About distance from home to the health facility was grouped into $<5$km and $>5$km away. Breast pain is grouped as feeling pain or not. Employee was grouped into employee or not employee.

## Search strategy

A systematic search of peer-reviewed, published literature in English was conducted to identify the factors contributing to the late presentation of breast cancer in East Africa (S2 File). We looked through the databases at PubMed, Hinari, EMBASE, Cochrane, CINHAL, Google Scholar, and Mednar to find pertinent research. To find pertinent key phrases, we first searched by article title in PubMed, Google, and Google Scholar. Secondly, we discovered related ideal keywords. Third, we conducted a second search using these phrases in the databases after looking for more research in the reference lists of all the recognized papers and publications. Terms like "breast cancer," "associated factors," "predictors," "determinants," "contributing factors," "prevalence," "magnitude," "proportion," "delayed patient presentation," "late presentation breast cancer," "late diagnosis breast cancer," "late diagnosis of the patient," "East Africa," In addition, eastern African countries, namely, Ethiopia, Ertriea, Sudan, South Sudan, Djibouti, Kenya, Rwanda, Zimbabwe, Tanzania, Uganda, Somalia, Burundi, Namibia, Botswana, Reunion, Mayotte, Seychelles, Madagascar, Marituis and Democratic republic of Congo were also included to ensure a comprehensive search. We experimented and improved utilizing several test searches, combining related search phrases with Boolean operators like OR and combining distinct notions using the Boolean operator AND.

## Data extraction

The data was extracted using Microsoft Excel. Two distinct data extraction formats were utilized to collect the information needed for analysis. In the extraction form for prevalence, we included the author's last name, the year the work was published, the study country, the study design, sample size, the frequency of breast cancer, the prevalence and its confidence interval,

and the quality score of each study. The author's last name and the year of publication were also included in the data extraction format for contributing factors. Every necessary piece of information was separately collected by two writers, who then cross-checked their findings and agreed on any discrepancies.

## Quality assessment/critical appraisal

The article was manually transferred to EndNote and checked for duplicates. The inclusion and exclusion criteria were applied to review the remaining articles, focusing on patient delay presentation of breast cancer in East Africa. The Newcastle-Ottawa quality appraisal checklist was used to evaluate the quality of individual studies [37] (S3 File).

Two reviewers evaluated each primary study individually, and a decision was made to accept or reject based on specific criteria. In case of disagreement, the average score of both reviewers was taken. A study was categorized as good quality if it scored more than 50% on quality assessment indicators. Each cross-sectional study was assessed using eight criteria: inclusion criteria, study subject and setting description, valid measurement of exposure, and identification of confounders using objective criteria, confounder handling strategies, outcome measurement, and statistical analysis. Eight cross-sectional studies met quality criteria and were included in the analysis.

## Result

A total of 1100 published studies (PubMed = 100, Hinari = 10, Cochrane Review = 85, EMBASE = 10, Google Scholar = 895) were identified. 200 duplicates were removed, leaving 900 abstracts for evaluation. 800 articles were excluded based on criteria methodological issues, not focused on East Africa and not relevant for breast cancer. Resulting in 100 articles was retained for full-text screening. 92 articles were further excluded for various reasons, leaving only 8 studies for final systematic and meta-analysis. Participant overlap was prevented by using the same data source, and studies were evaluated for quality before inclusion (Fig 1).

### Study characteristics

Only eight studies were included in this analysis [38–45]. Six articles were included from Ethiopia. Two articles were recruited from Rwanda and Sudan. From the included articles a study population of 2,842 participants, of whom 1095 participants delayed patient presentation of breast cancer. The included articles were published. All the included studies were facility based cross-sectional by design and reported delay presentation of breast cancer patients. The sample sizes across the studies ranged from63 [45] to 441 [42] (Table 2).

### Pooled prevalence of delayed patient presentation of breast cancer in East Africa

The overall pooled prevalence of delayed patient presentation of breast cancer was 61.85% (95% CI: 48.83–74.88%). Using the random effects model, the pooled effect size of delayed patient presentation of breast cancer showed statistically significant heterogeneity among the included studies (I2-98.1%, p<0001) (Fig 2).

### Subgroup analysis

We performed a subgroup analysis by country to address heterogeneity. The subgroup analysis showed that the prevalence of late presentation of breast cancer patients ranged from 50.0% in

## Identification of studies via database

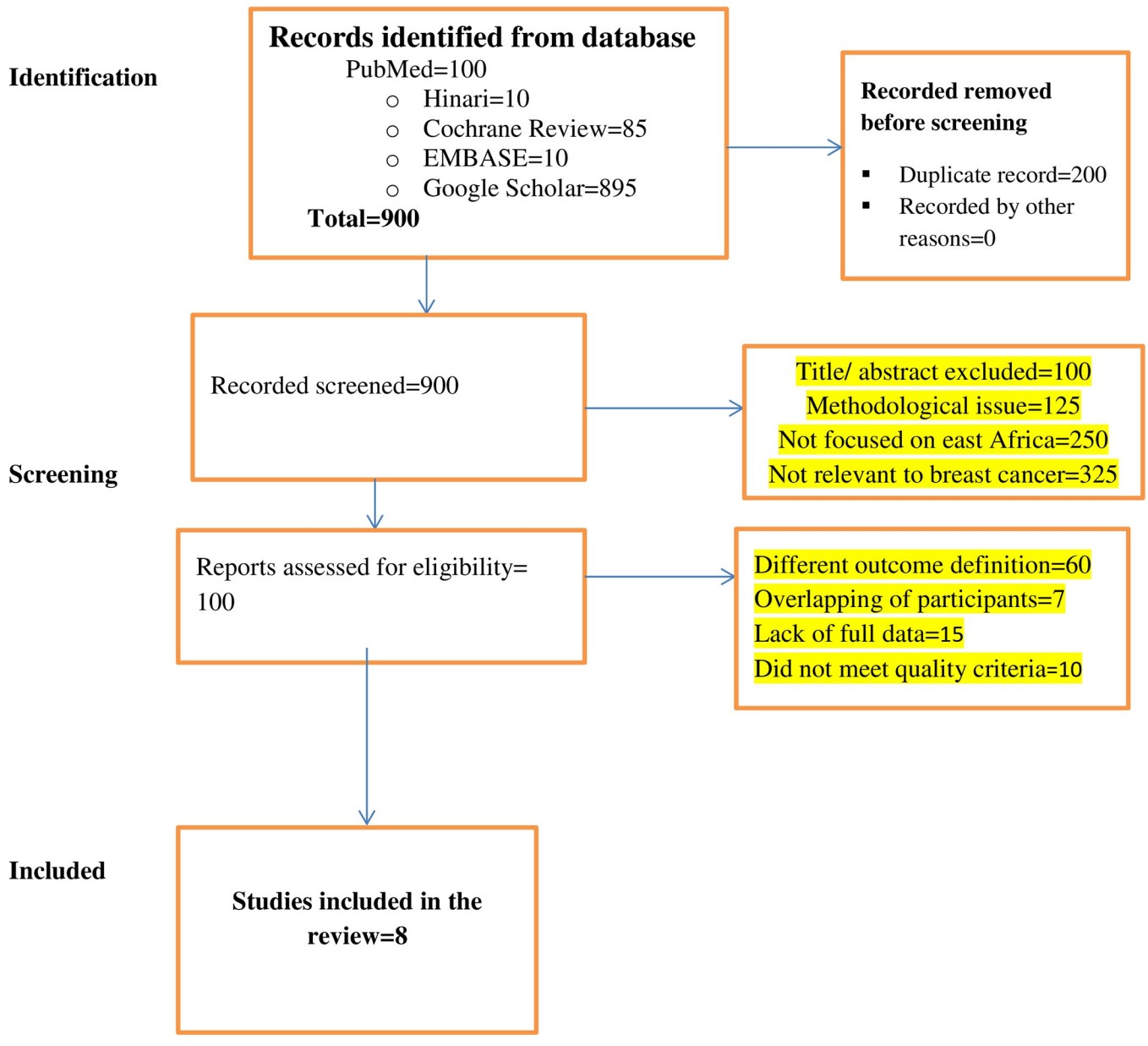

**Fig 1. PRISMA flowchart diagram of the study selection process.**

Rwanda to 74.6% in Sudan. Substantial heterogeneity was estimated, up to 98.6% in Ethiopia (Fig 3).

### Publication bias

Publication bias was viewed graphically by funnel plot asymmetry and tested through Egger's [46]. The p-value was >0.05; there was statistical evidence for the absence of publication bias using the Egger test. Egger's regression test was not statistically significant, with a P value of 0.089(Fig 4).

**Table 2. Description of included articles delayed presentation of breast cancer and associated factors in East Africa.**

| Id no | Author | Year | Study Design | Country | Actual Sample | Frequency | ES**[95%C] |
|---|---|---|---|---|---|---|---|
| 1 | Hassen AM, etal | 2021 | Cross-sectional | Ethiopia | 204 | 102 | 50.5(43.6, 57.4) |
| 2 | Tesfaw A, etal | 2020 | Cross-sectional | Ethiopia | 371 | 280 | 75.7(71.3, 80) |
| | Shewarega B, etal | 2023 | Cross-sectional | Ethiopia | 269 | 180 | 67(62.1, 71.7) |
| 4 | Muhammed JA, etal | 2022 | Cross-sectional | Ethiopia | 150 | 86 | 57.3(51.3, 63) |
| 5 | Gebremariam A, etal. | 2019 | Cross-sectional | Ethiopia | 441 | 159 | 36 (33, 38.7) |
| 6 | Abiye M, etal. | 2023 | Cross-sectional | Ethiopia | 206 | 157 | 76.7(70.8, 82.6) |
| 7 | Pace LE, etal | 2015 | Cross-sectional | Rwanda | 144 | 84 | 58 (51.9, 64.1) |
| 8 | Salih AM, etal | 2016 | Cross-sectional | Sudan | 63 | 47 | 74.6 (64.1, 85) |

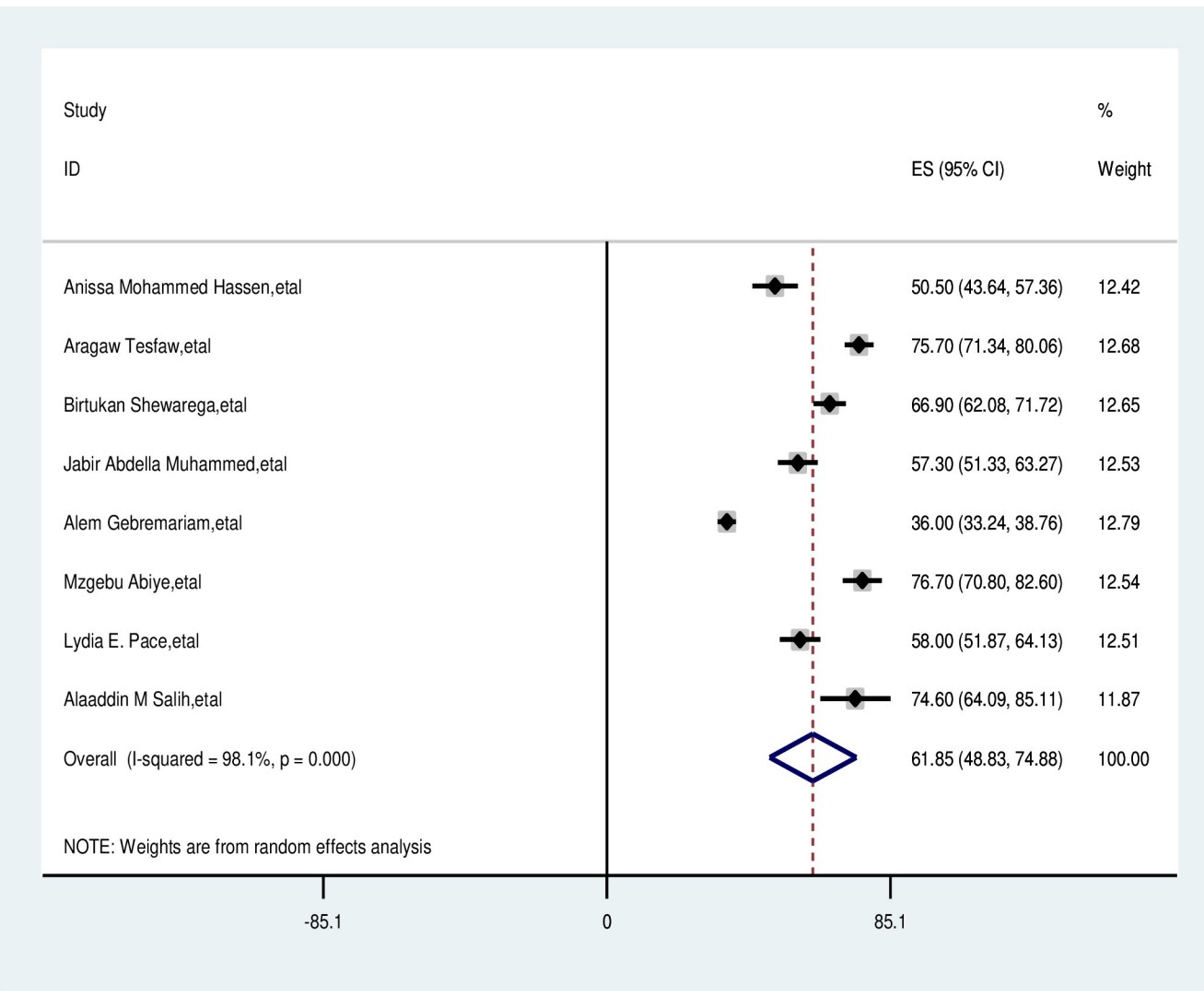

**Fig 2. Pooled prevalence late presentation of breast cancer patient in East Africa 2024.**

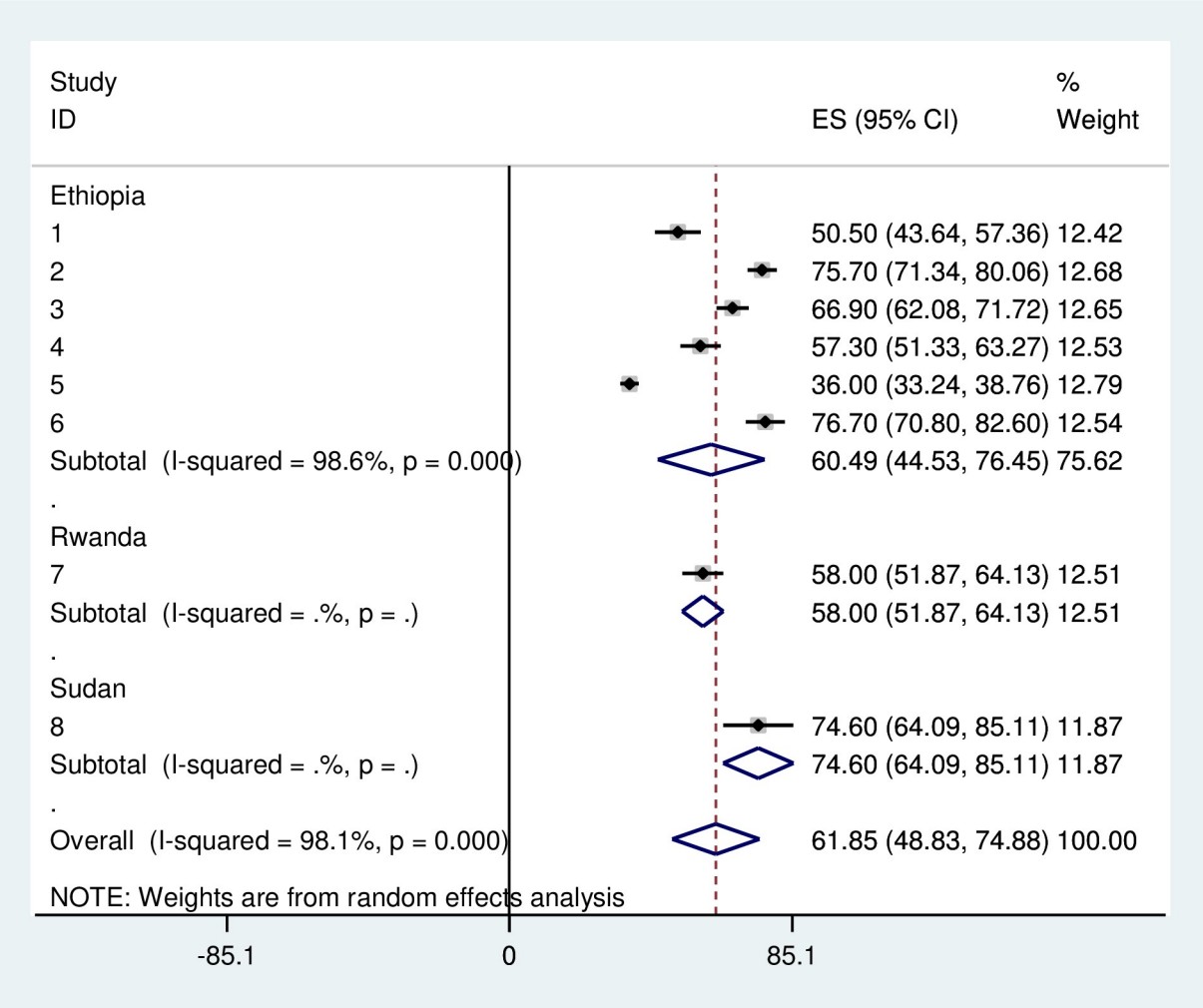

**Fig 3. Subgroup analysis of pooled prevalence late presentation of breast cancer patient in East Africa 2024.**

## Sensitivity analysis

In this meta-analysis, no single study dominated the pooled prevalence of delayed presentation of breast cancer patients in East Africa, according to the results of a random-effects model (Table 3).

**Table 3. Shows a sensitivity analysis of delayed presentation of breast cancer patients in east Africa.**

| Study omitted | Estimate | [95% Conf. Interval] | |
|---|---|---|---|
| Anissa Mohammed Hassen, et al | 63.47 | 48.89, | 78.04 |
| Aragaw Tesfaw, et al | 59.82 | 46.58, | 73.05 |
| Birtukan Shewarega, et al | 61.13 | 46.36, | 75.88 |
| Jabir Abdella Muhammed, et al | 62.51 | 47.67, | 77.34 |
| Alem Gebremariam, et al | 65.60 | 58.05, | 73.17 |
| Mezgebu Abiye, et al | 59.71 | 46.08, | 73.35 |
| Lydia E. Pace, et al | 62.41 | 47.64, | 77.18 |
| Alaaddin M Salih, et al | 60.13 | 46.23, | 74.03 |
| Combined | 61.85 | 48.82, | 74.87 |

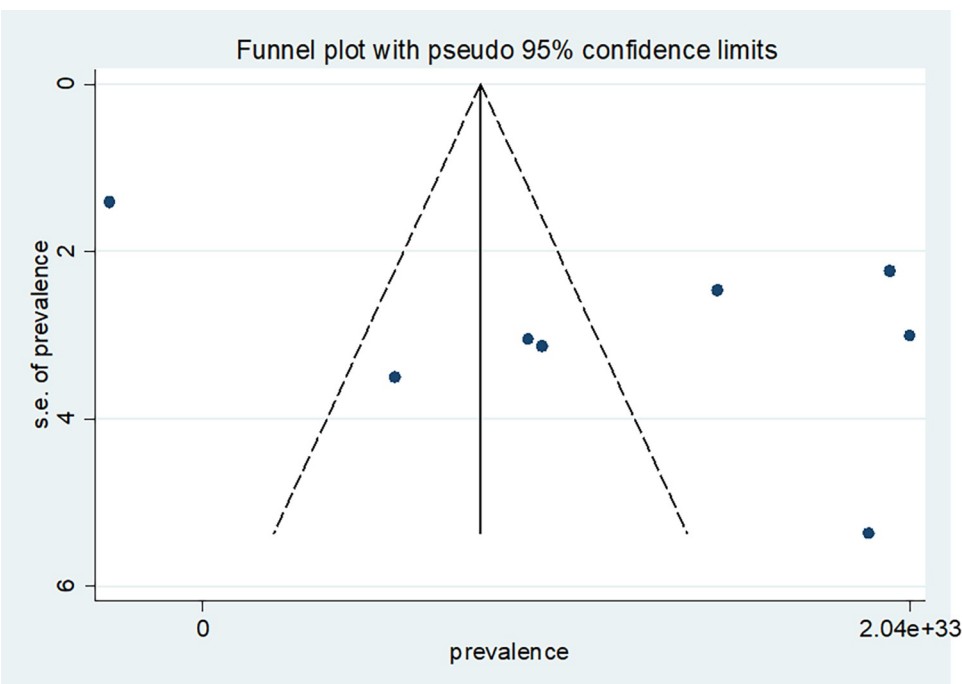

**Fig 4. Funnel plot publication bias plot for the prevalence of late presentation of breast cancer patient in East Africa.**

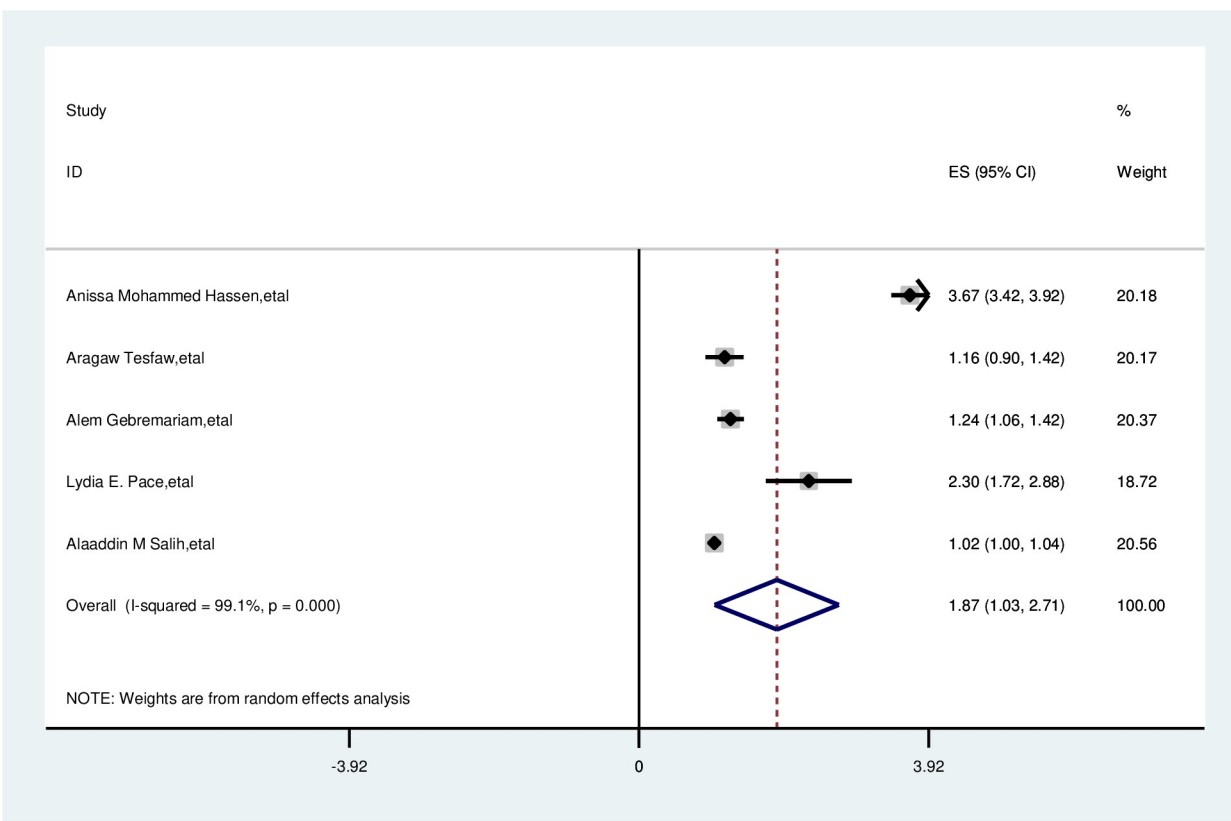

**Fig 5. Pooled odds ratio for the association between age with delay presentation of breast cancer patient in East Africa.**

**Table 4. Factors associated with delayed presentation of breast cancer patients in east Africa.**

| Variable | Exposed | Comparator | OR (95% CI) | I 2 |
|---|---|---|---|---|
| Visit Traditional medicine healer | Yes | No | 3.52; (1.43, 5.59) | 99.4% |
| Age | >40 | <40 | 1.87; (1.03, 2.71) | 99% |
| Educational status | Illiterate | Literate | 3.61; (2.39, 4.82) | 98.3% |
| Residence | Rural | Urban | 3.33; (2.16, 4.49) | 98.6% |
| Distance from home to health facility | >5km | <5km | 2.89; (1.54, 4.24) | 97.5% |
| Breast Pain | Not feeling breast pain | Feeling breast pain | 2.42; (1.09, 3.74) | 99% |

### Factors associated with delay presentation of breast cancer patient in East Africa

We included 11 selected variables to identify relationships with the delayed presentation of breast cancer patients in East Africa. Of these, six variables, namely age = >40 years, low educational level, rural residence, visit traditional healer, distance>5km away from the health facility and not feeling breast pain were significantly associated with delayed presentation of breast cancer patients (Table 4). The review also demonstrated that employee; marital status, not having a lump under the armpit, not having awareness of breast cancer and no family history of breast cancer had no statistically significant association with delayed presentation of breast cancer.

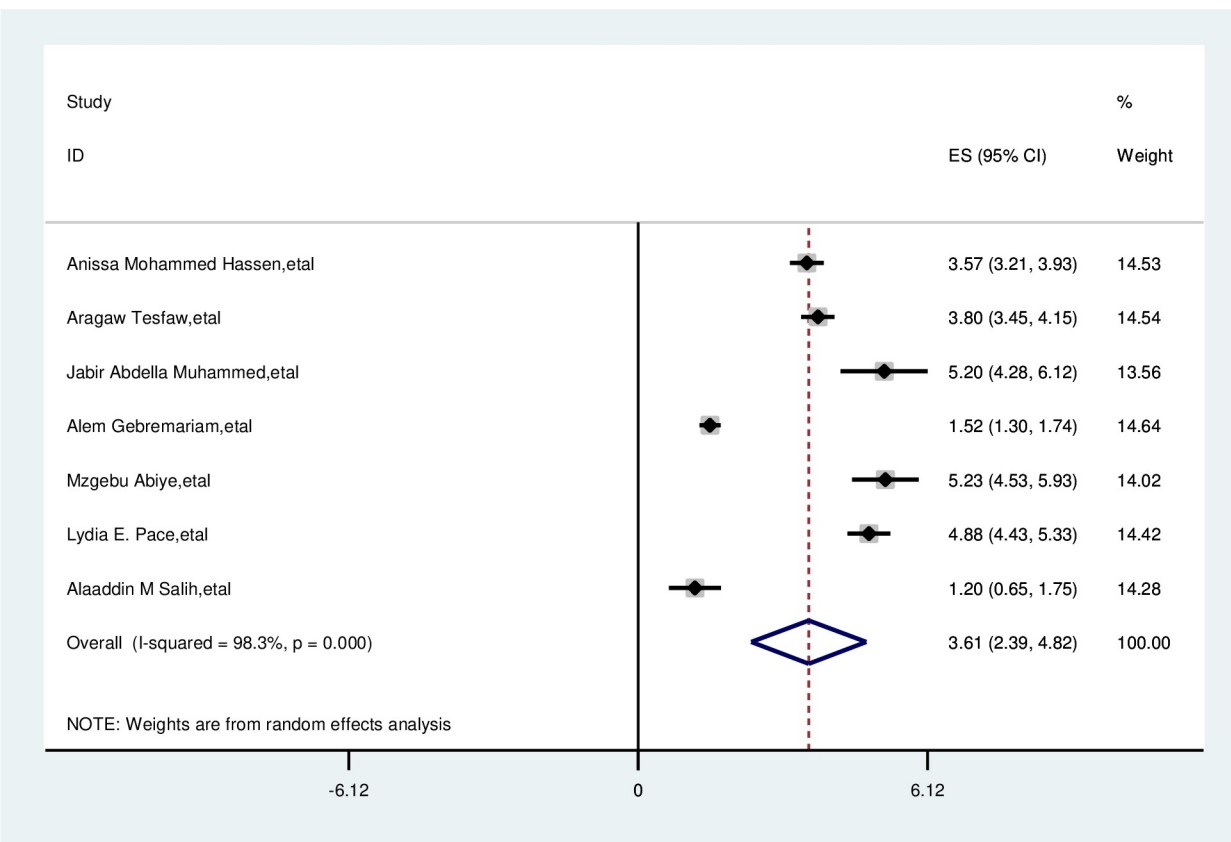

**Fig 6. Pooled odds ratio for the association between educational status with delay presentation of breast cancer patient in East Africa.**

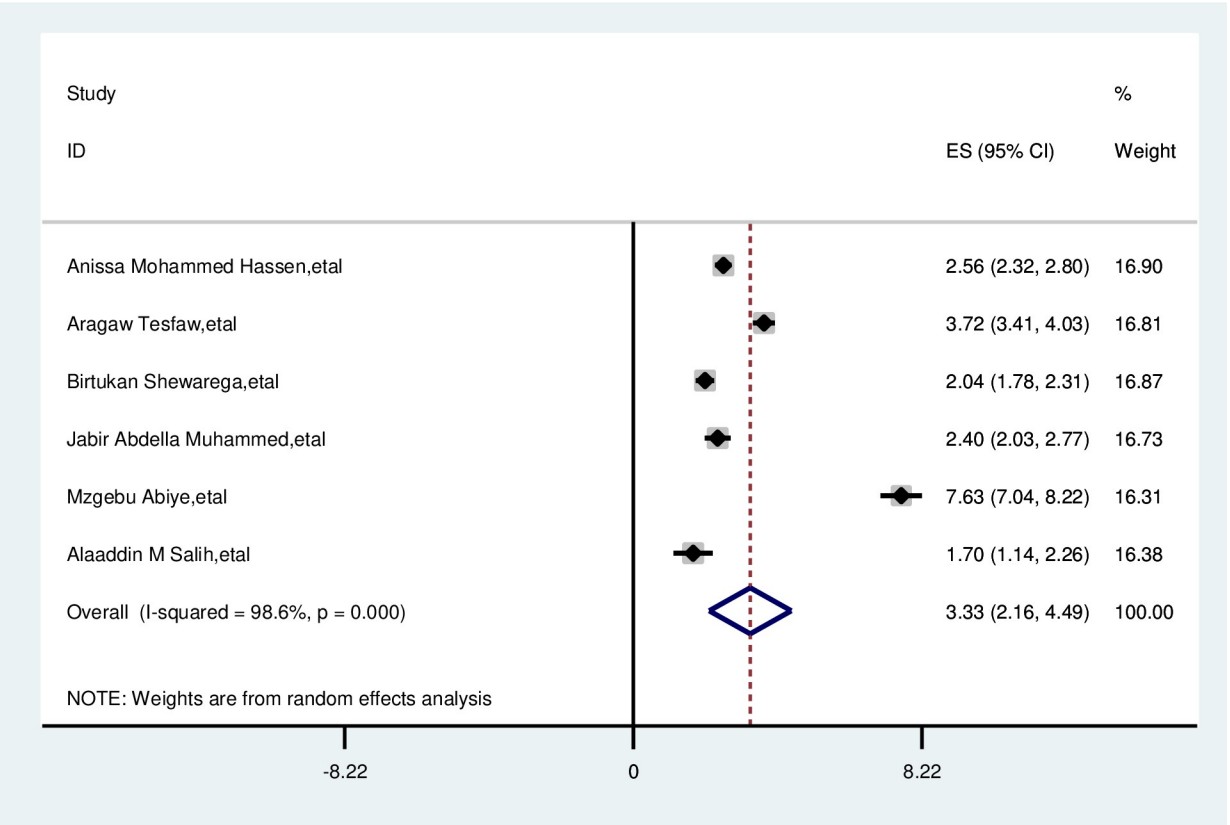

**Fig 7. Pooled odds ratio for the association between rural residences with delay presentation of breast cancer participants in East Africa.**

**Age of participant.** In the overall analysis of this study age of the participant was significantly associated with delay in presentation of breast cancer patients. Participants with age = >40 years were nearly 2 times more likely to delay presentation than those of age less than 40 years (OR, 1.87; 95% CI: 1.03, 2.71). A random effects model was assumed for the analysis as I2 (99.1%) and Egger test 0.220 with a p-value of (<0.001) showed statistically significant heterogeneity among the included studies for this factor analysis (Fig 5).

**Educational status.** The overall analysis of studies showed education had a positive impact on delayed breast cancer patients. Participants with low educational status were 3.6 times more likely delay than their counterparts (OR, 3.61; 95% CI: 2.39, 4.82). A random effects model was assumed for the analysis as I2 (98.3%) and Egger test 0.118 with a p-value of (<0.001) showed statistically significant heterogeneity among the included studies for this factor analysis (Fig 6).

**Residence.** The overall analysis of studies showed that residence had a positive association with delayed presentation of breast cancer patients. Participants with rural residence were 3.3 times more likely to delay than that of their counterparts (OR, 3.33; 95% CI: 2.16, 4.49). A random effects model was assumed for the analysis as I2 (98.6%) and Egger test 0.307 with a p-value of (<0.001) showed statistically significant heterogeneity among the included studies for this factor analysis (Fig 7).

**Absence of breast pain.** The overall analysis of studies showed not having breast pain had a positive impact on delayed breast cancer patients. Participants who had no breast pain were 2.4 times more likely delay than their counterparts (OR, 2.42; 95% CI: 1.09, 3.74). A random

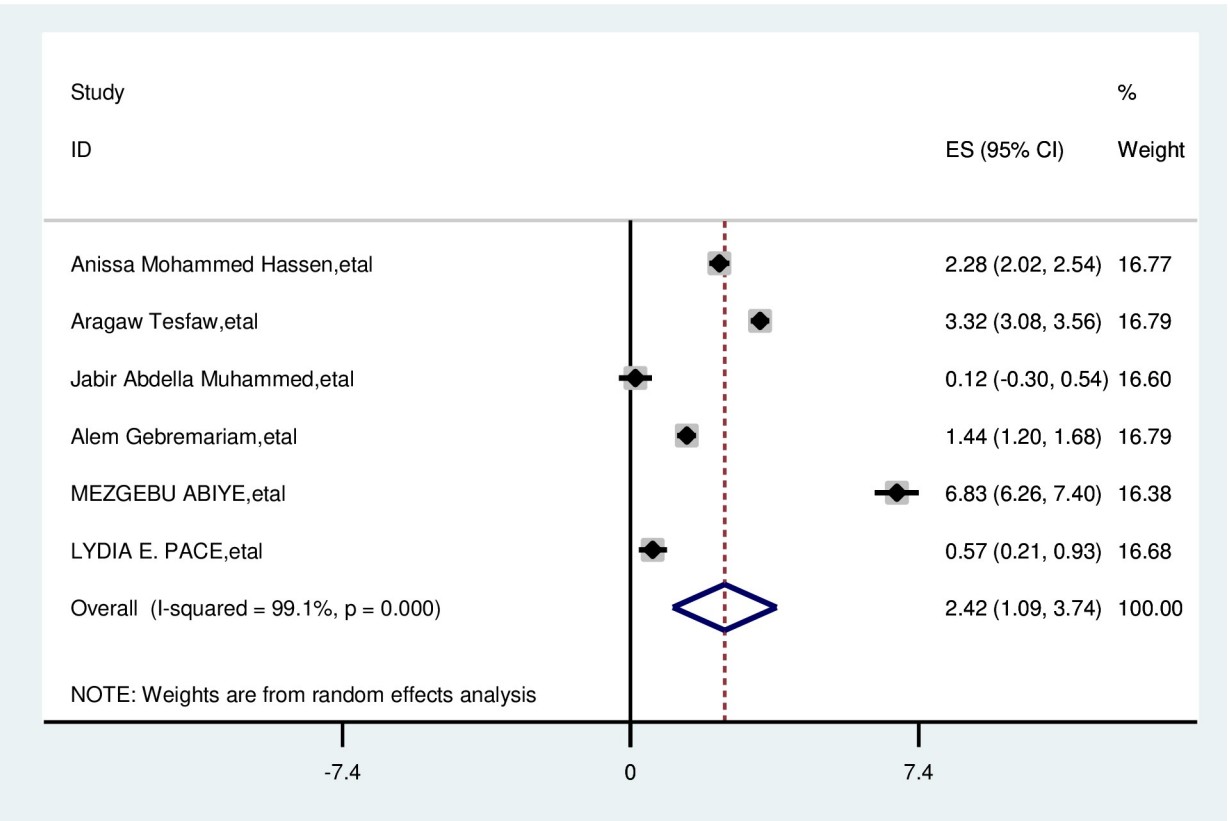

**Fig 8. Pooled odds ratio for the association between not having breast pain with delay presentation of breast cancer participants in East Africa.**

effects model was assumed for the analysis as I2 (99.1%) and Egger test 0.801 with a p-value of (<0.001) showed statistically significant heterogeneity among the included studies for this factor analysis (Fig 8).

**Distance from health facility.** The overall analysis of studies showed the distance from the health facility had a positive impact on delayed breast cancer patients. Participants who were>5km away from the health facility were nearly 3 times more likely delayed than that their counterparts (OR, 2.89; 95% CI: 1.54,4.24). A random effects model was assumed for the analysis as I2 (97.5%) and Egger test 0.081 with a p-value of (<0.001) showed statistically significant heterogeneity among the included studies for this factor analysis (Fig 9).

**Visit traditional healer.** The overall analysis of studies showed that visiting traditional healers had a positive impact on delayed breast cancer patients. Participants who have visited traditional healers were 3.52 times more likely delayed than that of their counterparts (OR, 3.52; 95% CI 1.43, 5.59). A random effects model was assumed for the analysis as I2 (99.4%) and Egger test 0.345 with a p-value of (<0.001) showed statistically significant heterogeneity among the included studies for this factor analysis (Fig 10).

## Factors not associated with delayed presentation of a breast cancer patient in East Africa

**Employee.** The meta-analysis showed that employed participants were not significantly associated with delayed presentation. The overall odds ratio was 0.63 with a 95% CI of 0.11–

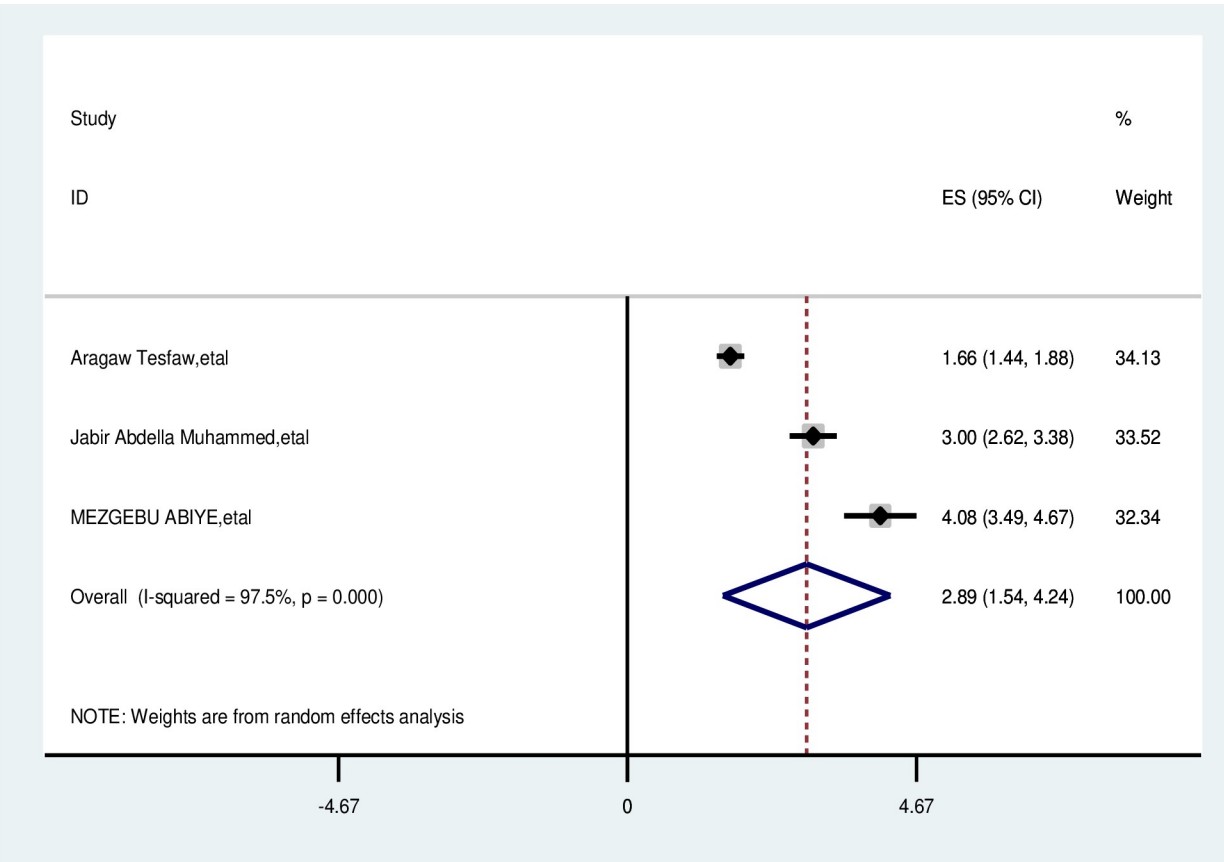

**Fig 9. Pooled odds ratio for the association between >5km away from health facility with delay presentation of breast cancer participants in East Africa.**

1.15, and a p-value of <0.021. A random effects model was used for the analysis, showing statistically significant heterogeneity among the included studies (I2 = 74.4%, p-value <0.021) as depicted in Fig 11.

**Marital status.** Marital status was not associated with delayed presentation of breast cancer patients. The overall odd ratio of married women is 1.39(0.90–1.88). Random effect model was used I2,93%) and the Egger test was 0.294 with significant heterogeneity for this analysis (Fig 12).

**Not having awareness of breast cancer.** The overall analysis of studies showed that not having awareness of breast cancer has no association with delayed presentation of breast cancer patients (OR, 1.58; 95% CI: 0.69, 2.46). A random effects model was assumed for the analysis as I2 (97.2%) and Egger test 0.618 with a p-value of (<0.001) showed statistically significant heterogeneity among the included studies for this factor analysis (Fig 13).

**Not having lump under armpit.** The overall analysis of studies showed no lump under the armpit had no positive impact on delayed breast cancer patients (OR, 3.34; 95% CI: 0.30, 6.38). A random effects model was assumed for the analysis as I2 (99.7%) and Egger test 0.801 with a p-value of (<0.001) showed statistically significant heterogeneity among the included studies for this factor analysis (Fig 14).

**Have no family history of breast cancer.** The overall analysis of studies showed no family history of breast cancer had no positive effect on delayed breast cancer patients (OR, 2.693; 95% CI: 0.268,5.118). A random effects model was assumed for the analysis as I2 (99.7%) and

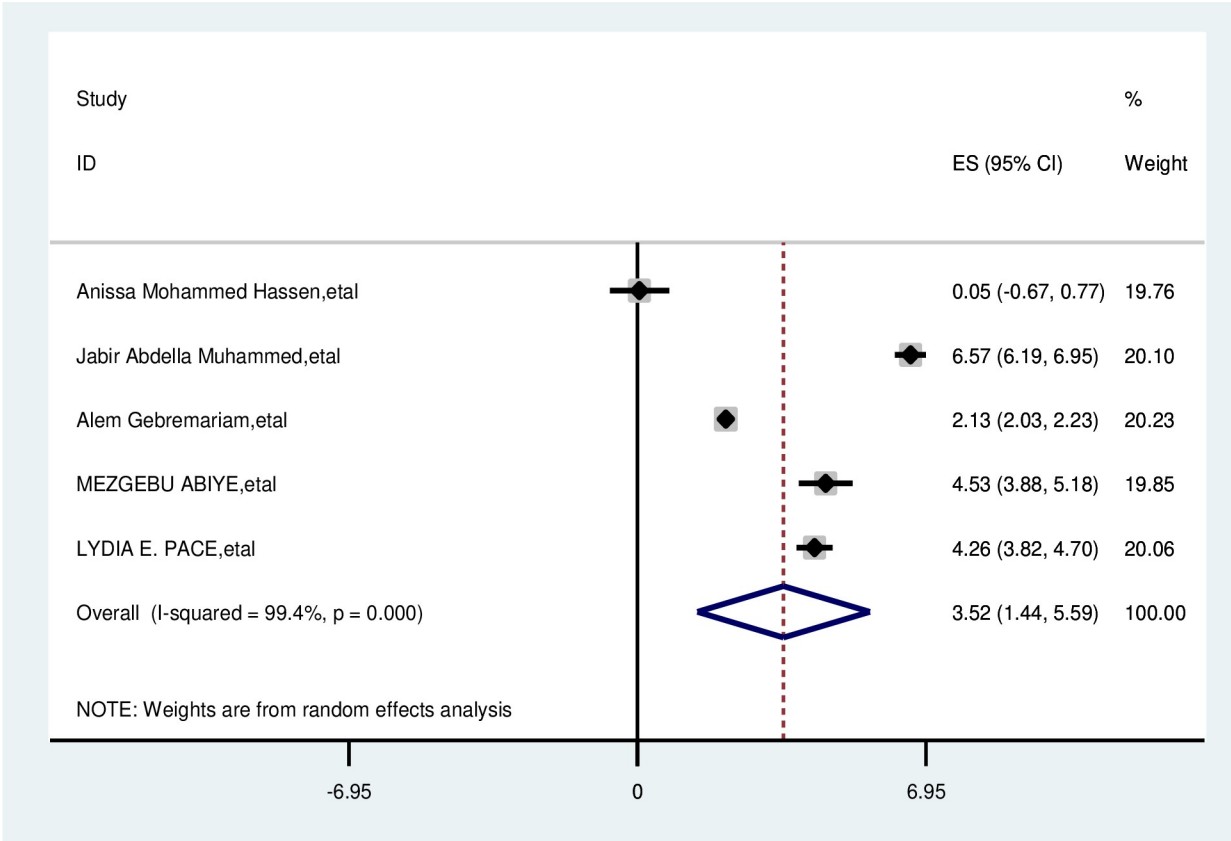

**Fig 10. Pooled odds ratio for the association between visit traditional healer with delay presentation of breast cancer in East Africa.**

Egger test 0.611 with a p-value of (<0.001) showed statistically significant heterogeneity among analyzed factors (Fig 15).

## Discussion

The study conducts a detailed analysis of the coverage of delayed patient presentations for breast cancer in Eastern Africa. Breast cancer patients who delayed presenting were shown to have a pooled overall prevalence of 61.85% (95% CI: 48.83–74.88%). This high prevalence of delayed presentation of breast cancer has serious implications for public health in East Africa. There might be increased morbidity and mortality, increased economic burden for treatment and exacerbating poverty, and reduced quality of life due to more severe symptoms. Around the world, between 25% and 89% of breast cancer patients present after their disease has progressed. This disparity might be caused by several factors, including access to the health care facility, quality of health care service, awareness and education level, socioeconomic status, cultural factors, quality of the research, health insurance, financial barriers, the availability of medical supplies, and the article's publication date.

This study is consistent with previous studies done in different countries in indenesia58% [47],inNigeria72% [48], in Nigeria 68% [49], a systematic review and meta-analysis done in Africa 54% [11], in India50% [50], and in Saudi60.7% [31]. There were a few potential causes, including comparable social levels, cultural beliefs, identical educational backgrounds, and comparable health-seeking habits. However, it was higher than previous research done in

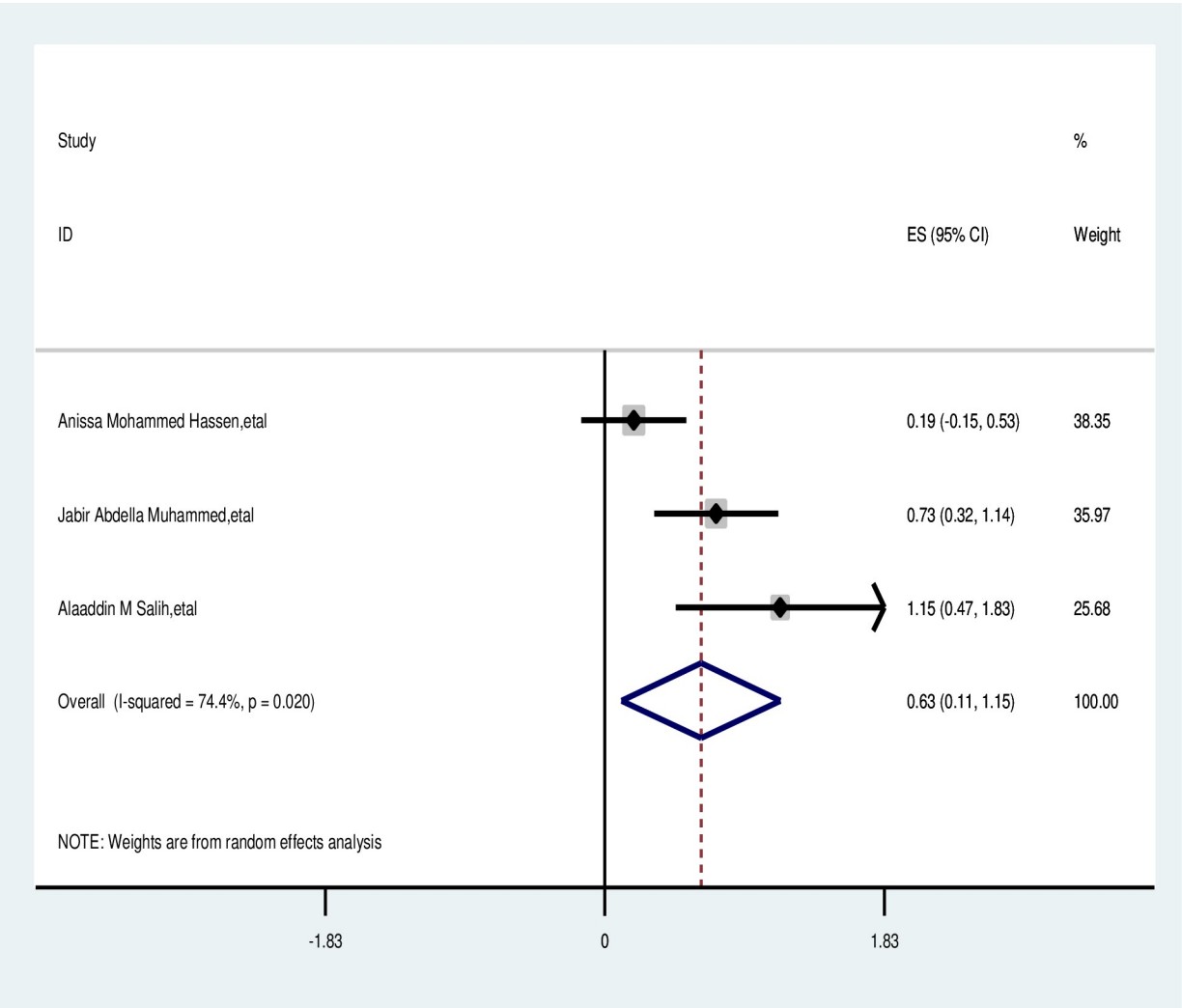

**Fig 11. Pooled odds ratio for the association between employed with delay presentation of breast cancer in East Africa.**

Iraq44% [51], in Indonesia 43.4% [47], in Pakistan39% [52], in pakistan43.8% [34], and in Iran31.7% [33]. This high percentage can be explained by socio-demographic factors or by weak or inefficient measures implemented by the competent healthcare authorities in East African regions. Those regions are unstable as a result of natural disasters and conflict, and they also have poor accessibility and availability of health-related services. It might be the poor state of many health systems in East Africa and their declining capacities to lead cancer preventive initiatives and respond to the overall health needs of the population, as compared to developed countries, are also major concerns [53, 54].

This systematic review and meta-analysis revealed significant heterogeneity among the included studies. The variation is due to data collection methods, measurement of delayed presentation breast cancer, sample size representativeness, regional difference, cultural difference, and quality of the included variables. Addressing these inconsistencies will enhance the comparability of the studies and improve the reliability of the conclusion drawn from this meta-analysis.

The subgroup analysis of the study between countries indicated that the highest level of delayed presentation of breast cancer was in Sudan (74.6%). This is higher than a systematic

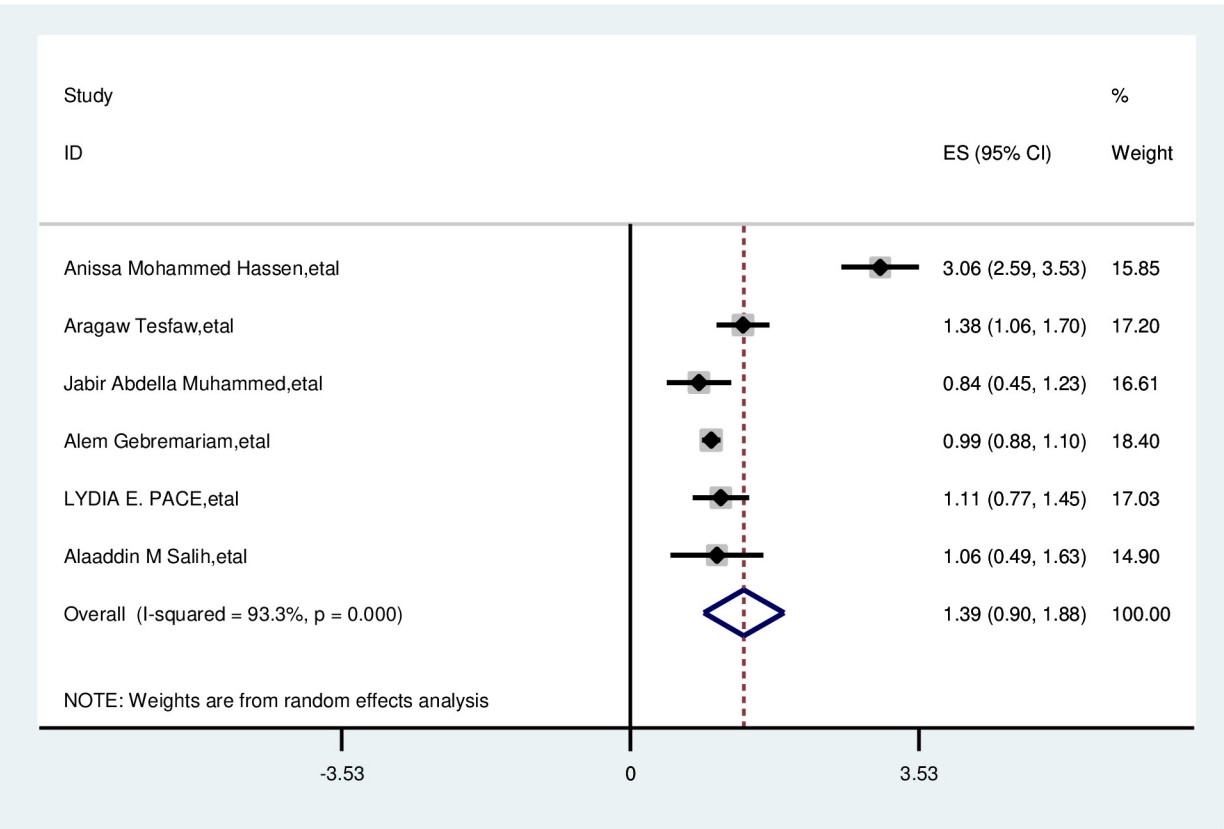

**Fig 12. Pooled odds ratio for the association between married with delay presentation of breast cancer in East Africa.**

review and meta-analysis done in Africa54% [11].The possible explanation might be a low level of awareness about health-seeking behavior, socio-cultural factors, small sample size, and study design differences.

In this meta-analysis the contributing factors were the age of more than40 years, illiterate/low level of education, rural residence,>5km away from the health facility, visiting traditional healer and not feeling breast pain were significantly associated with delayed presentation of a breast cancer patient.

This review found that those age>40 years were more likely to delay the presentation of breast cancer than their counterparts. This is consistent with a study done in Pakistan [52], in the United Kingdom [55], in Saudi [31], in Nigeria [49], in the Middle East [19] and Estonia [56].The possible reason was that age advances decreased health-seeking behavior and prioritized other health issues over breast cancer screening and treatment. There may be a lack of autonomy and fear of sharing the problems with health care provider [30–32].However, it contradicts earlier studies [33, 57], because older patients may be anxious about various comorbidities and quickly referred to health institution [58].

These review findings show that illiterate/low level of education breast cancer patients had a stronger association with delayed presentation than educated ones. This is similar to prior studies done in Iraq [51], systematic reviews, and meta-analyses done in different countries [10, 18–21]and in Pakistan [32, 52, 57]. The potential explanation of low education is not being aware of the symptoms of breast problems, the belief that symptoms go away by themselves [59], the severity of the disease, and cultural intervention. More if the participant is not

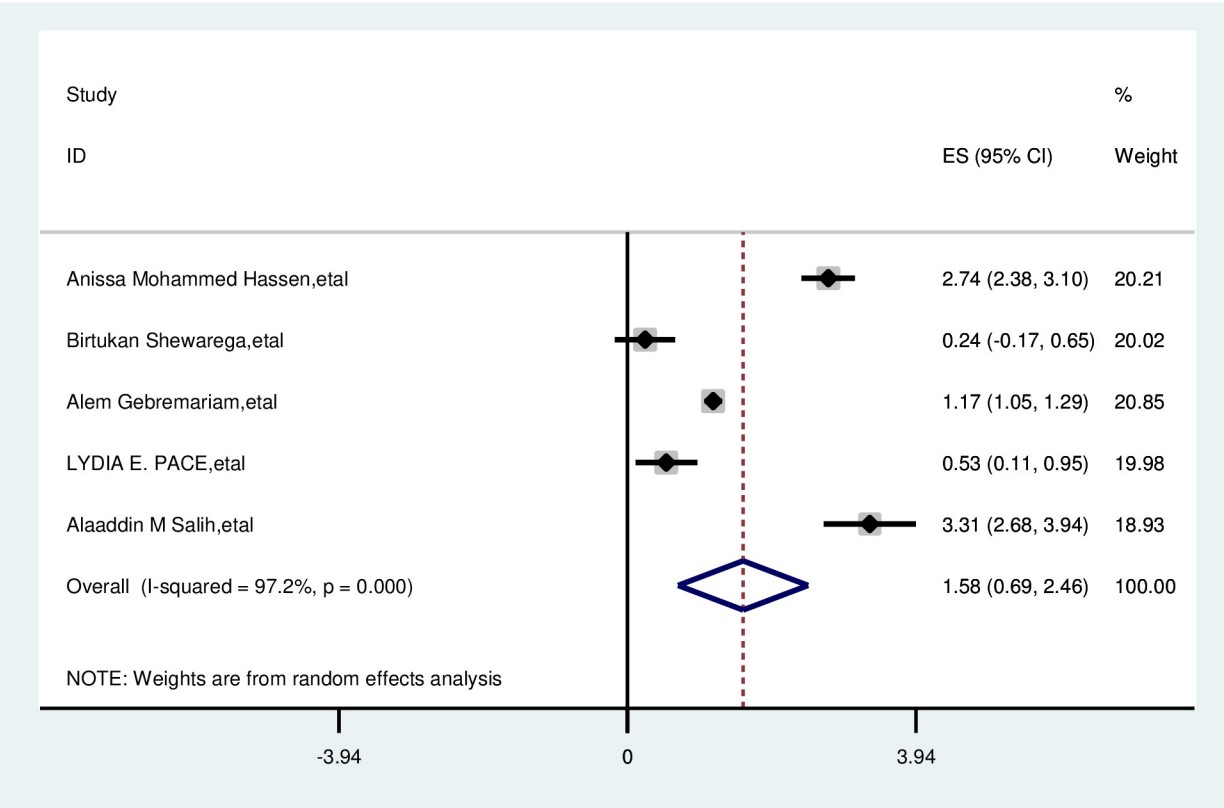

**Fig 13. Pooled odds ratio for the association between no awareness of breast cancer with delay presentation of breast cancer in East Africa.**

educated less likely to disclose the symptoms to friends and health care providers as a result of the delay of presentation of breast cancer. Additionally, higher education levels might increase the likelihood of comprehending the health campaign messages with terminology that is not representative of local dialects.

Moreover, this meta-analysis found that rural residence had a strong association with delayed presentation of breast cancer. These figures are similar to a study done in Saudi [31] and in Pakistan [57].The justification is the rural part of East Africa region had a low level of awareness about health-seeking consultation and poor healthcare services(accessibility, availability) [29, 30, 34] and they are not near for information. Plus, women who come from rural countries may have difficulty with transportation to a nearby health center, and traveling a long distance to get an appropriate diagnosis, which in turn may result in delayed presentation.

This meta-analysis also found that not feeling/absence of breast pain was more likely to delay the presentation of breast cancer than that of counterparts. This is similar to a study done in Iraq [51], in Estonia [56], in Palestinian [60], and in Saudi [31].The possible reason was patient their breast is asymptomatic they can't appreciate the disease and they believe that mass without pain is normal physiology and resolves spontaneously. This is also correlated with breast self-examination is important for early detection and decreasing delayed presentation [61].

Furthermore, the review found that participants who visit traditional healers had a strong association with delayed presentation of breast cancer. This is supported by prior studies [11, 21–23].The explanation was they perceived traditional healers were more curative than

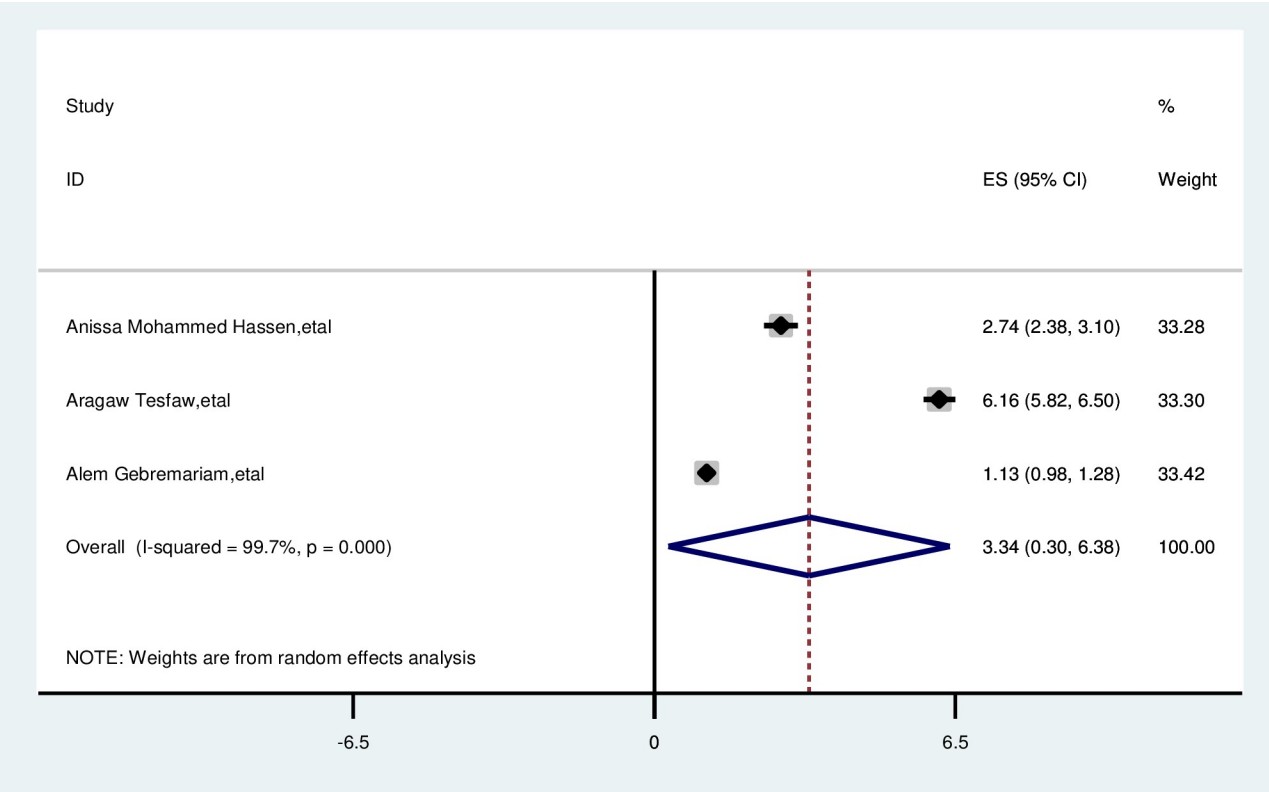

**Fig 14. Pooled odds ratio for the association between no lump underarmpit with delay presentation of breast cancer in East Africa.**

modern medicine and patients wasted their time with traditional medicine. In addition, because of fear of surgery of the breast and false belief [30–32], they are abstaining from the health institution for health care services. So, while taking those remedies, most patients delay coming to the health facility leading to worsening of symptoms and advanced stage.

The final significant factor for this research found that distance >5km from the health facility was a significant factor with delay presentation of breast cancer patients. It has been reported that East African women living far from health institutions are particularly vulnerable to late presentation breast cancer, partially due to the high cost of transportation [30], low socio-economic status [18, 33], transportation problem [30] and waste time by long journeys to reach health care facilities.

## Strength of the study

Strengths of this study include our rigorous review of existing published literature. An assessment of study quality and heterogeneity, providing insight into the reliability of the findings and compressive synthesis of evidence.

## Limitations of the study

One of the limitations of the study is the fact that recall bias may not have been eliminated from the study, as almost all studies included in the study were cross-sectional and it is possible that the outcome variable was under- or overestimated. The findings of this meta-analysis should be done with due consideration of the substantial heterogeneity between included studies. The high heterogeneity indicates that the studies are not sufficiently similar to combine

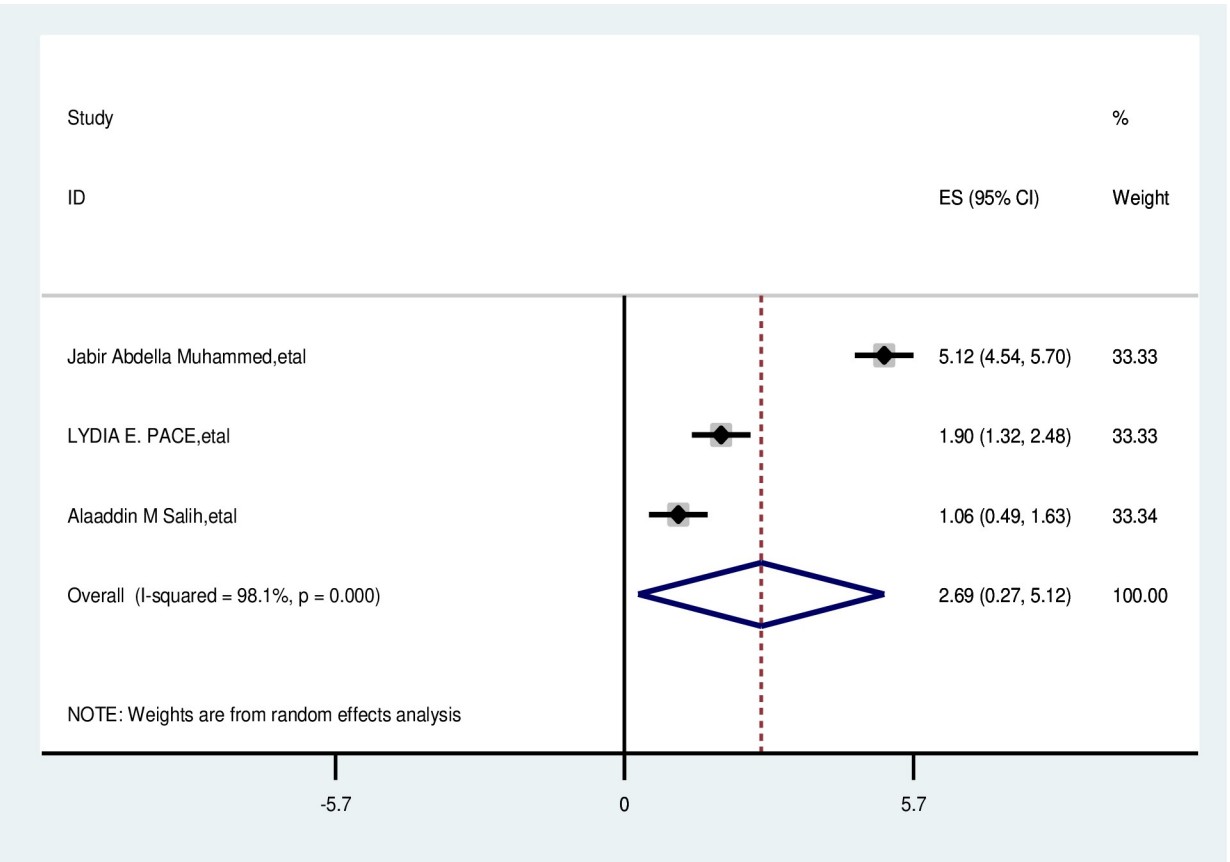

**Fig 15. Pooled odds ratio for the association between have no family history of breast cancer with delay presentation of breast cancer in East Africa.**

meaningfully, leading to potentially misleading conclusions. Studies that were published only in the English language were taken into account.

We suggest for future researchers the inclusion of diverse study designs, focus on subgroup analysis, and improved statistical techniques for handling heterogeneity and missing data, it is better to take a qualitative approach and also include articles published in different languages.

## Conclusion

This meta-analysis found that over half of breast cancer patients in East Africa experience delays in the presentation of breast cancer. Significant factors associated with delayed presentation include age over 40 years, illiteracy, rural residence, use of traditional healers, distance greater than 5 km from a health facility, and absence of breast pain. These results highlight the critical need for targeted public health interventions to address these barriers. Healthcare stakeholders and policymakers focus on enhancing awareness, improving education, and increasing healthcare access are essential to reduce delays and improve early detection and treatment of breast cancer in East Africa.

## Supporting information

**S1 File. The PRISMA checklist.**
(DOCX)

**S2 File. Newcastle-Ottawa quality assessment scale for cross-sectional studies used in the systematic review and meta-analysis 2024.**
(DOCX)

**S3 File. Searching strategy.**
(DOCX)

**S4 File. Description of inclusion/exclusion articles.**
(DOCX)

**S5 File. Data extraction of included articles in the systematic review and meta- analysis.**
(DOCX)

**S6 File. Characteristics of the included articles.**
(DOCX)

## Acknowledgments

We acknowledge Woldia University for the provision of internet service to conduct this meta-analysis.

## Author Contributions

**Conceptualization:** Betelhem Ejigu.

**Formal analysis:** Betelhem Ejigu.

**Methodology:** Tadele Emagneneh.

**Software:** Getinet Kumie.

**Supervision:** Getinet Kumie.

**Visualization:** Tadele Emagneneh, Abebaw Alamrew.

**Writing – original draft:** Chalie Mulugeta, Abebaw Alamrew.

**Writing – review & editing:** Chalie Mulugeta.

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
