## [Decision Letter · Decision Letter 0]

7 Jul 2024

PONE-D-24-14681Delayed Presentation of Breast Cancer Patients and Contributing Factors in East Africa: Systematic Review and Meta- Analysis.PLOS ONE

Dear Dr. Mulugeta,

Thank you for submitting your manuscript to PLOS ONE. After careful consideration, we feel that it has merit but does not fully meet PLOS ONE’s publication criteria as it currently stands. Therefore, we invite you to submit a revised version of the manuscript that addresses the points raised during the review process. Please submit your revised manuscript by Aug 21 2024 11:59PM. If you will need more time than this to complete your revisions, please reply to this message or contact the journal office at plosone@plos.org. Please include the following items when submitting your revised manuscript:A rebuttal letter that responds to each point raised by the academic editor and reviewer(s). You should upload this letter as a separate file labeled 'Response to Reviewers'.A marked-up copy of your manuscript that highlights changes made to the original version. You should upload this as a separate file labeled 'Revised Manuscript with Track Changes'.An unmarked version of your revised paper without tracked changes. You should upload this as a separate file labeled 'Manuscript'.

We look forward to receiving your revised manuscript.

Kind regards,

Wenjie Shi

Academic Editor

PLOS ONE

Journal Requirements:

2. Please amend your list of authors on the manuscript to ensure that each author is linked to an affiliation. Authors’ affiliations should reflect the institution where the work was done (if authors moved subsequently, you can also list the new affiliation stating “current affiliation:….” as necessary).

Additional Editor Comments:

Please revise the manuscript according to the reviewers' comments.

Reviewers' comments:

Reviewer's Responses to Questions

**Comments to the Author**

1. Is the manuscript technically sound, and do the data support the conclusions?

Reviewer #1: Partly

Reviewer #2: Yes

Reviewer #3: Yes

Reviewer #4: Yes

2. Has the statistical analysis been performed appropriately and rigorously? 

Reviewer #1: Yes

Reviewer #2: Yes

Reviewer #3: Yes

Reviewer #4: Yes

3. Have the authors made all data underlying the findings in their manuscript fully available?

Reviewer #1: Yes

Reviewer #2: Yes

Reviewer #3: Yes

Reviewer #4: Yes

4. Is the manuscript presented in an intelligible fashion and written in standard English?

Reviewer #1: Yes

Reviewer #2: Yes

Reviewer #3: Yes

Reviewer #4: Yes

5. Review Comments to the Author

Reviewer #1: This study, a systematic review and meta-analysis, examined the prevalence and contributing factors of delayed presentation among breast cancer patients in East Africa and found that more than half of patients experienced delayed diagnosis, which was significantly associated with factors such as age, education level, place of residence and visits to traditional healers. However, there are several issues that need to be addressed before the paper can be considered for publication.

1) Does the study cover all countries in East Africa or is it limited to certain countries?

2) Does the study include the most recent data or is it limited to a specific time period?

3) Did the study consider the impact of socioeconomic status on the behaviour of delayed medical seeking?

4) Did the study examine the effect of access to health care on delayed medical consultation?

5) Did the study analyse the impact of culture and beliefs on the behaviour of delayed medical consultation?

6) Did the study consider the impact of the diagnostic capabilities of different health care facilities on delayed diagnosis?

7) Did the study assess the impact of patients' knowledge about breast cancer on their behaviour in seeking medical care?

8) Does the study consider the impact of patients' self-perception of symptoms on their health care seeking behaviour?

9) The manuscript should be proofread for grammatical errors, spelling mistakes or unclear presentation.

10) The methods section should be more precise, detailed and scientific.

Reviewer #2: General Comments:

The manuscript presents a systematic review and meta-analysis on the delayed presentation of breast cancer patients in East Africa, identifying key factors contributing to late presentation. This is a significant and timely topic, given the high mortality associated with breast cancer in the region. The study follows a rigorous methodological approach and provides valuable insights. However, several areas require major revisions to improve the clarity, robustness, and overall quality of the manuscript.

Major Revisions:

1. Abstract Clarity and Conciseness:

o The abstract should be more concise and clearly highlight the key findings and their implications. It currently includes repetitive information and lacks a clear structure.

o The results section in the abstract should provide specific statistics on delay prevalence and the most significant factors contributing to the delay, rather than a general overview.

2. Introduction:

o The introduction section is lengthy and contains excessive background information. Condense it to focus more on the rationale for the study, specific research questions, and the study's significance.

o Some of the statistics and details can be moved to a background subsection if necessary.

3. Methods:

o Search Strategy: The search strategy section is comprehensive but lacks clarity in describing the inclusion and exclusion criteria. Consider using a table to summarize these criteria for better readability.

o Quality Assessment: More detail is needed on how the Newcastle-Ottawa Scale was applied and the criteria used for assessing the quality of included studies. A table summarizing the quality scores of the included studies would be helpful.

4. Results:

o The results section should present the key findings more clearly, with distinct subsections for different aspects of the study (e.g., prevalence of delayed presentation, factors contributing to the delay).

5. Discussion:

o The discussion should provide a deeper analysis of the results, comparing them with findings from other regions or previous studies. Discuss the implications of the high prevalence of delayed presentation and the identified factors in more detail.

o Highlight the strengths and limitations of the study more explicitly. Discuss how the limitations might have affected the results and suggest areas for future research.

6. Conclusion:

o The conclusion should be more focused on summarizing the key findings and their implications for public health policies and interventions in East Africa.

o Avoid introducing new information in the conclusion section.

7. Overall Language and Style:

o The manuscript would benefit from a thorough proofreading to correct grammatical errors and improve the overall readability. Some sentences are overly complex and could be simplified.

Specific Comments:

1. Abstract:

o "The overall analysis of delay presentation breast cancer patient was 61.85%..." should be rephrased for clarity.

o Results should be summarized with specific data points and concise explanations.

2. Introduction:

o Condense the first two paragraphs to avoid redundancy. Focus on the unique context of East Africa.

o The last paragraph of the introduction should clearly state the study objectives and research questions.

3. Methods:

o Clarify the search terms and databases used. Consider presenting the search strategy in a table.

o Provide a rationale for the selection of studies based on the Newcastle-Ottawa Scale.

4. Results:

o The PRISMA flow diagram should be included to illustrate the study selection process.

o Present the findings related to each factor contributing to the delay in separate subsections or a summary table.

5. Discussion:

o Compare the study’s findings with those of similar studies from other regions.

o Discuss potential reasons for the high prevalence of delayed presentation in East Africa.

6. Conclusion:

o Emphasize the need for targeted interventions and policy changes based on the study’s findings.

o Suggest practical steps for healthcare providers and policymakers to reduce delayed presentations.

Conclusion:

The manuscript addresses an important public health issue and provides valuable data on the delayed presentation of breast cancer patients in East Africa. However, significant revisions are necessary to improve the clarity, structure, and depth of the analysis. With these improvements, the manuscript has the potential to make a substantial contribution to the field.

Reviewer #3: Reviewer Comments:

The manuscript titled "Factors Contributing to Late Presentation of Breast Cancer in East Africa: A Systematic Review and Meta-Analysis" presents a comprehensive investigation into a critical public health issue. The study is timely, given the rising global incidence and mortality rates of breast cancer, particularly in less developed regions like East Africa.

1. The introduction effectively contextualizes the significance of the study within the broader global health landscape. It clearly articulates the urgency of addressing late presentations of breast cancer in East Africa, which is essential for guiding future interventions and policies.

2.The authors adhered to the PRISMA guidelines meticulously, ensuring transparency and reproducibility in their systematic review and meta-analysis. The search strategy was robust, encompassing multiple databases and employing comprehensive search terms relevant to the study objectives.

3.The analysis of data regarding the prevalence of late patient presentations and associated factors was thorough and well-documented. The inclusion of quality assessment criteria and the justification for study selection criteria added to the credibility of the findings.

4.The discussion effectively synthesizes the findings, providing insights into the implications for breast cancer management and public health strategies in East Africa. It appropriately discusses the limitations of the study and suggests avenues for future research.

5. The character of the included studies needs to be described in detail.

The manuscript is generally well-written and organized, though some sections could benefit from minor revisions for clarity and flow. Attention to enhancing the coherence between sections and refining language to improve readability would further strengthen the manuscript.

Reviewer #4: The authors reported about the delayed presentation of breast cancer patients and contributing factors in East Africa.

We need to know if there any updated data after 2020 in Background section.

And it is necessary to explain the heterogeneity in the Discussion section.

6. PLOS authors have the option to publish the peer review history of their article (what does this mean?). If published, this will include your full peer review and any attached files.

Reviewer #1: **Yes: **Xiaodong Zou

Reviewer #2: **Yes: **Yan Li

Reviewer #3: No

Reviewer #4: No

---

## [Author Response · Author response to Decision Letter 0]

28 Jul 2024

Thank you dear editors and reviewers for your valuable comments. Based on your comment we submit the response entitled by Response to reviewers in the submissions system. The responses to the reviewers' comments and suggestions have been highlighted in blue and the main document of the manuscript in yellow for better assessment.

---

## [Decision Letter · Decision Letter 1]

7 Aug 2024

PONE-D-24-14681R1Delayed Presentation of Breast Cancer Patients and Contributing Factors in East Africa: Systematic Review and Meta- Analysis.PLOS ONE

Dear Dr. Mulugeta,

Thank you for submitting your manuscript to PLOS ONE. After careful consideration, we feel that it has merit but does not fully meet PLOS ONE’s publication criteria as it currently stands. Therefore, we invite you to submit a revised version of the manuscript that addresses the points raised during the review process. Please submit your revised manuscript by Sep 21 2024 11:59PM. If you will need more time than this to complete your revisions, please reply to this message or contact the journal office at plosone@plos.org. Please include the following items when submitting your revised manuscript:A rebuttal letter that responds to each point raised by the academic editor and reviewer(s). You should upload this letter as a separate file labeled 'Response to Reviewers'.A marked-up copy of your manuscript that highlights changes made to the original version. You should upload this as a separate file labeled 'Revised Manuscript with Track Changes'.An unmarked version of your revised paper without tracked changes. You should upload this as a separate file labeled 'Manuscript'.If applicable, we recommend that you deposit your laboratory protocols in protocols.io to enhance the reproducibility of your results. Protocols.io assigns your protocol its own identifier (DOI) so that it can be cited independently in the future. For instructions see: https://journals.plos.org/plosone/s/submission-guidelines#loc-laboratory-protocols. Additionally, PLOS ONE offers an option for publishing peer-reviewed Lab Protocol articles, which describe protocols hosted on protocols.io. Read more information on sharing protocols at https://plos.org/protocols?utm_medium=editorial-email&utm_source=authorletters&utm_campaign=protocols.

We look forward to receiving your revised manuscript.

Kind regards,

Wenjie Shi

Academic Editor

PLOS ONE

Journal Requirements:

Additional Editor Comments:

please make revisions according to Review 2's comments

Reviewers' comments:

Reviewer's Responses to Questions

**Comments to the Author**

1. If the authors have adequately addressed your comments raised in a previous round of review and you feel that this manuscript is now acceptable for publication, you may indicate that here to bypass the “Comments to the Author” section, enter your conflict of interest statement in the “Confidential to Editor” section, and submit your "Accept" recommendation.

Reviewer #1: All comments have been addressed

Reviewer #2: All comments have been addressed

Reviewer #3: All comments have been addressed

Reviewer #4: All comments have been addressed

2. Is the manuscript technically sound, and do the data support the conclusions?

Reviewer #1: Yes

Reviewer #2: Yes

Reviewer #3: Yes

Reviewer #4: Yes

3. Has the statistical analysis been performed appropriately and rigorously? 

Reviewer #1: Yes

Reviewer #2: Yes

Reviewer #3: Yes

Reviewer #4: Yes

4. Have the authors made all data underlying the findings in their manuscript fully available?

Reviewer #1: Yes

Reviewer #2: Yes

Reviewer #3: Yes

Reviewer #4: Yes

5. Is the manuscript presented in an intelligible fashion and written in standard English?

Reviewer #1: Yes

Reviewer #2: Yes

Reviewer #3: Yes

Reviewer #4: Yes

6. Review Comments to the Author

Reviewer #1: In the revised manuscript, the authors have addressed all my concerns in a very convincing manner. As such I support the publication of this original article in PLOS ONE.

Reviewer #2: Overall Evaluation

The paper provides a comprehensive analysis of the potential for using mRNA expression data to predict biochemical recurrence (BCR) in prostate cancer (PCa) patients pre-operatively. The use of machine learning methodologies to model time-to-event data is innovative and demonstrates significant improvements in predictive performance compared to traditional clinical models. The study is well-structured and the methodology is sound, however, there are several areas where the paper could benefit from additional clarification and minor revisions.

Major Strengths

1. Innovative Approach: The use of mRNA expression data pre-operatively for BCR prediction in PCa is novel and holds promise for improving patient outcomes.

2. Comprehensive Methodology: The paper employs a range of machine learning models and provides a thorough evaluation of their performance using multiple metrics.

3. Clinical Relevance: The focus on pre-operative prediction aligns well with the clinical need for early and accurate decision-making in prostate cancer treatment.

Minor Revisions

1. Clarification of Calibration Methodology:

o The paper describes two forms of calibration analyses but the explanation of the calibration curves and how they were derived is somewhat dense. Consider breaking down the steps more clearly, perhaps with additional visual aids or flowcharts to guide the reader through the process.

o Specifically, provide more detailed explanations on how the Kaplan-Meier estimates were used in the calibration curves and the rationale behind using the quintiles.

2. Detailed Explanation of Feature Selection:

o While the study discusses the feature selection rates, there is limited discussion on why certain mRNA variables were selected over others. A deeper analysis of the biological relevance of the frequently selected mRNA variables (e.g., DNAH8, ABCC11, ESM1) and their known roles in PCa or other cancers would strengthen the discussion.

o It would be beneficial to include a table summarizing the key mRNA variables, their known functions, and any previous associations with cancer to provide context to their selection.

3. Discussion of Model Limitations:

o The paper acknowledges the modest cohort size and single-centre data source as limitations. It would be useful to discuss potential biases that might arise from this and how they might impact the generalizability of the results.

o Consider elaborating on how multi-centre validation could address these limitations and the specific steps that will be taken in future research to ensure robustness and applicability across diverse populations.

4. Figure Improvements:

o Figures 1 and 2 are crucial for understanding the calibration performance of the models but could benefit from higher resolution and clearer labels. Ensure that all axes and legends are easy to read.

o In Figure 5, it would be helpful to add more context or annotations to highlight key observations regarding the expression levels of mRNA variables in patients receiving neoadjunctive therapy versus those who did not.

5. ROC and DCA Analysis:

o The ROC and DCA analyses are well-presented but could be further enhanced by including confidence intervals for the AUC values in the ROC plots directly, rather than just in the table. This visual representation would make it easier to compare the performance across models.

o In the DCA plots, consider adding a brief explanation of how net benefit is interpreted and its clinical implications, as some readers may not be familiar with this analysis.

6. Discussion on Clinical Implementation:

o The conclusion mentions the potential for clinical integration of these models but does not elaborate on the practical steps required to achieve this. Provide more details on what would be needed for these models to be adopted in a clinical setting, including any regulatory considerations, necessary validation studies, and potential barriers to implementation.

Conclusion

This paper presents a significant contribution to the field of prostate cancer research by demonstrating the potential of mRNA-based pre-operative prediction models. With minor revisions to enhance clarity, provide deeper biological insights, and discuss implementation strategies, this study could offer valuable guidance for the development of precision medicine tools in oncology.

Reviewer #3: (No Response)

Reviewer #4: (No Response)

7. PLOS authors have the option to publish the peer review history of their article (what does this mean?). If published, this will include your full peer review and any attached files.

Reviewer #1: **Yes: **Xiaodong Zou

Reviewer #2: **Yes: **Yan Li

Reviewer #3: No

Reviewer #4: No

---

## [Author Response · Author response to Decision Letter 1]

12 Aug 2024

Dear reviewer great thanks for your continuous support for the improvement of our manuscript. The response of our manuscript was sent by the system.

---

## [Decision Letter · Decision Letter 2]

20 Aug 2024

Delayed Presentation of Breast Cancer Patients and Contributing Factors in East Africa: Systematic Review and Meta- Analysis.

PONE-D-24-14681R2

Dear Dr. Mulugeta,

We’re pleased to inform you that your manuscript has been judged scientifically suitable for publication and will be formally accepted for publication once it meets all outstanding technical requirements.

Kind regards,

Wenjie Shi

Academic Editor

PLOS ONE

Additional Editor Comments (optional):

Reviewers' comments:

Reviewer's Responses to Questions

**Comments to the Author**

1. If the authors have adequately addressed your comments raised in a previous round of review and you feel that this manuscript is now acceptable for publication, you may indicate that here to bypass the “Comments to the Author” section, enter your conflict of interest statement in the “Confidential to Editor” section, and submit your "Accept" recommendation.

Reviewer #2: All comments have been addressed

2. Is the manuscript technically sound, and do the data support the conclusions?

Reviewer #2: Yes

3. Has the statistical analysis been performed appropriately and rigorously? 

Reviewer #2: Yes

4. Have the authors made all data underlying the findings in their manuscript fully available?

Reviewer #2: Yes

5. Is the manuscript presented in an intelligible fashion and written in standard English?

Reviewer #2: Yes

6. Review Comments to the Author

Reviewer #2: The revised manuscript presents a well-conducted systematic review and meta-analysis on the delayed presentation of breast cancer patients in Eastern Africa. The study provides a comprehensive evaluation of the prevalence of delayed presentation and identifies key factors associated with these delays. Overall, the manuscript is methodologically sound and offers valuable insights into a critical public health issue. I recommend accepting the manuscript for publication with minor revisions to address the following specific comments.

7. PLOS authors have the option to publish the peer review history of their article (what does this mean?). If published, this will include your full peer review and any attached files.

Reviewer #2: **Yes: **Yan Li

---

## [Editor Report · Acceptance letter]

4 Sep 2024

PONE-D-24-14681R2 

PLOS ONE

Dear Dr. Mulugeta, 

I'm pleased to inform you that your manuscript has been deemed suitable for publication in PLOS ONE. Congratulations! Your manuscript is now being handed over to our production team.

Kind regards, 

on behalf of

Prof. Wenjie Shi 

Academic Editor

PLOS ONE